# General Analysis of Heat Shock Factors in the *Cymbidium ensifolium* Genome Provided Insights into Their Evolution and Special Roles with Response to Temperature

**DOI:** 10.3390/ijms25021002

**Published:** 2024-01-13

**Authors:** Ruiyue Zheng, Jiemin Chen, Yukun Peng, Xuanyi Zhu, Muqi Niu, Xiuming Chen, Kai Xie, Ruiliu Huang, Suying Zhan, Qiuli Su, Mingli Shen, Donghui Peng, Sagheer Ahmad, Kai Zhao, Zhong-Jian Liu, Yuzhen Zhou

**Affiliations:** 1Ornamental Plant Germplasm Resources Innovation & Engineering Application Research Center, Key Laboratory of National Forestry and Grassland Administration for Orchid Conservation and Utilization, College of Landscape Architecture and Art, Fujian Agriculture and Forestry University, Fuzhou 350002, China; fafuruiyue@163.com (R.Z.); c2549539102@163.com (J.C.); pyk20001022@163.com (Y.P.); zxy316377506@163.com (X.Z.); marveln@163.com (M.N.); cxm291802145@163.com (X.C.); xiekai_0526@163.com (K.X.); huangruiliu06@163.com (R.H.); jamaisvuzhan@163.com (S.Z.); sqljya@163.com (Q.S.); fjpdh@fafu.edu.cn (D.P.); sagheerhortii@gmail.com (S.A.); 2College of Life Sciences, Fujian Normal University, Fuzhou 350117, China; paramountshen@163.com (M.S.); zhaokai@fjnu.edu.cn (K.Z.)

**Keywords:** orchid, cis-elements, gene expression, abiotic stress, floral scent regulation

## Abstract

Heat shock factors (*HSF*s) are the key regulators of heat stress responses and play pivotal roles in tissue development and the temperature-induced regulation of secondary metabolites. In order to elucidate the roles of *HSF*s in *Cymbidium ensifolium*, we conducted a genome-wide identification of *CeHSF* genes and predicted their functions based on their structural features and splicing patterns. Our results revealed 22 *HSF* family members, with each gene containing more than one intron. According to phylogenetic analysis, 59.1% of HSFs were grouped into the A subfamily, while subfamily HSFC contained only two HSFs. And the *HSF* gene families were differentiated evolutionarily between plant species. Two tandem repeats were found on Chr02, and two segmental duplication pairs were observed on Chr12, Chr17, and Chr19; this provided evidence for whole-genome duplication (WGD) events in *C. ensifolium*. The core region of the promoter in most *CeHSF* genes contained cis-acting elements such as AP2/ERF and bHLH, which were associated with plant growth, development, and stress responses. Except for *CeHSF11*, *14*, and *19*, each of the remaining *CeHSF*s contained at least one miRNA binding site. This included binding sites for miR156, miR393, and miR319, which were responsive to temperature and other stresses. The *HSF* gene family exhibited significant tissue specificity in both vegetative and floral organs of *C. ensifolium*. *CeHSF13* and *CeHSF15* showed relatively significant expression in flowers compared to other genes. During flower development, CeHSF15 exhibited markedly elevated expression in the early stages of flower opening, implicating critical regulatory functions in organ development and floral scent-related regulations. During the poikilothermic treatment, *CeHSF14* was upregulated over 200-fold after 6 h of heat treatment. *CeHSF13* and *CeHSF14* showed the highest expression at 6 h of low temperature, while the expression of *CeHSF15* and *CeHSF21* continuously decreased at a low temperature. The expression patterns of *CeHSFs* further confirmed their role in responding to temperature stress. Our study may help reveal the important roles of *HSFs* in plant development and metabolic regulation and show insight for the further molecular design breeding of *C. ensifolium*.

## 1. Introduction

*Cymbidium ensifolium* belongs to the orchid genus *Cymbidium* and holds high ornamental and medicinal values [1,2,3]. It serves as a potential parental line for the modern horticultural hybridization of orchids, especially in the development of heat-tolerant, early-flowering, and fragrant hybrids [4]. In recent years, abiotic stresses such as drought, high temperatures, salinity, and rising carbon dioxide have affected plant growth and agricultural development [5,6,7]. Among these, high temperature is one of the biggest challenges faced worldwide [8]. In the past few decades, there has been an increase in regions experiencing extremely high temperatures in China. This trend is primarily concentrated in the Yangtze River Delta region, eastern Sichuan, the Beijing–Tianjin–Hebei region, and southeastern coastal areas [9]. Fujian Province, the origin of *C. ensifolium*, has experienced dramatic temperature changes and frequent heat events in recent decades [10]. Due to various abiotic stresses, the habitats of *C. ensifolium* are declining, leading to the shrinking of wild populations [11,12]. *C. ensifolium*, currently domesticated as vital ornamental orchids, has formed a global industry chain, requiring large quantities to meet market demands. Gaining a clear understanding of the heat response mechanisms of *C. ensifolium* is one of the key methods in modern orchid breeding. However, there has been a scarcity of relevant research.

Heat stress transcription factors (HSFs) are crucial in the plant stress response, playing pivotal roles in adapting to various abiotic stresses. During biotic or abiotic stresses, TFs bind to specific promoter regions of target genes, activating or repressing their transcription and orchestrating defensive responses [13]. For instance, they regulate the expression of stress-responsive genes like heat shock proteins (HSPs) [14]. HSFs possess five conserved domains: the N-terminal DNA-binding domain (DBD), oligomerization domain (OD), nuclear localization signal (NLS), nuclear export signal (NES), and C-terminal activation domain (CAD) [15,16]. The CAD can modulate gene expression in response to heat shock [17]. The NLS and NES regulate the nuclear import/export and distribution of HSPs. Due to the conservation of DBD and OD, they are often used to identify HSFs. They also enable binding to HSP promoters [18]. Based on structural features, HSFs are classified into three subfamilies: HSFA, HSFB, and HSFC [19]. These HSF subfamilies facilitate plant responses to abiotic stress by activating HSP expression [20].

Current studies have shown that *HSFs* respond to heat stress signals in plants and regulate growth and development [21]. In *Arabidopsis*, salt and osmotic and cold stress can increase the expression of the stress-responsive gene *HSFA6b* via ABA mediation [22]. *HSFA* was found to modulate thermotolerance and cell wall integrity in *Aspergillus fumigatus* [23]. Overexpression of *HSFA9* in *Helianthus annuus* promoted the accumulation of carotenoids, chlorophyll, and chloroplasts, enhancing cotyledon development [24]. Under drought, heat, and salt stresses, *TrHSFB2a* acted as a negative regulator of stress tolerance in white clovers [25]. The *HSFB* of *Marchantia polymorpha* played a pivotal role in thermal reactions, overseeing the development of meristem branching and fostering the formation of anther stems [26]. *AtHSFB4* overexpression caused shortened root length in *Arabidopsis*, impacting root development [27]. Furthermore, the *HSF* gene exhibited a certain correlation with the formation of floral compounds in specific plants, affecting the flower development process [28,29,30]. Hence, exploring the functions of *HSF* genes in enhancing both plant stress resistance and breeding holds paramount significance.

Since the identification of the *HSF* family in *A. thaliana* [31], increasing numbers of *HSF* families have been uncovered in other plants, including tomato [20], *Oryza sativa* [32], *Secale cereale* [33], *Zea mays* [34], *Pisum sativum* [35], kiwifruit [36], *Salvia miltiorrhiza* [37], and *Glycine max* [38]. However, little documentation exists on this family in orchids. During the previous researches of the research group, We found that this protein has multiple functions, especially for the formation of endogenous floral substance. Here, we performed a whole identification of *CeHSF* genes in *C. ensifolium*. Through analyzing their phylogeny, gene structure, motif composition, cis elements, protein–protein interactions, miRNA targets, and expression patterns, we obtained a panoramic regulation and induction model. We also examined the expression of *CeHSFs* under heat and cold treatments using qRT-PCR to find out the detailed gene reponses for further usage. Our findings may establish a foundation for functional studies of *HSFs* in orchids.

## 2. Results

### 2.1. Phylogenetic Analysis and Identification of HSFs in C. ensifolium

Based on the genome data of *C. ensifolium*, a total of 22 *HSF* genes were identified (Appendix A). A total of 21 AtHSFs, 25 OsHSFs, 18 PeHSFs, and 22 CeHSFs were used to construct a phylogenetic tree. Phylogenetic analysis showed that the 22 members were divided into three groups (Group A, Group B, and Group C) (Figure 1). This phylogenetic classification was consistent with the presence of conserved DNA-binding domains (DBDs) and the established classification of *Arabidopsis* HSFs. Group A had the most *C. ensifolium* HSF members with thirteen, while Group C had the fewest with only two. There were seven *C. ensifolium* HSF members in Group B. The phylogenetic tree results highlighted some differences between monocot and eudicot HSFs, such as eudicots having one clade in Group C (C1), while monocots had two (C1 and C2).

The *CeHSFs* were designated as *CeHSF1* to *CeHSF22* according to their chromosomal sequences and locations. Corresponding gene IDs, names, chromosomal loci, etc., were tabulated (Table 1). Analysis of various physicochemical parameters showed that the CeHSF proteins ranged from 190 to 514 amino acids in length, 22.4 to 51.6 kDa in molecular weight, 4.82 to 9.61 in theoretical pI, 39.1 to 72.6 in instability index, 66.7 to 83.3 in aliphatic index, and −0.899 to −0.396 in grand average hydropathicity, consistent with hydrophilic proteins (Table 1).

### 2.2. Gene Structures and Conserved Motifs of C. ensifolium HSFs

Based on genomic DNA sequences, the gene structures and conserved motifs of *CeHSFs* in *C. ensifolium* were analyzed. HSF proteins all contained the DBD domain (three α-helices and four β-folded layers), which was highly conserved. Sequence alignment between *C. ensifolium* and *Arabidopsis* indicated that *C. ensifolium* HSF proteins had the typical DBD structure (Figure 2A). In the α3-helix, all AtHSFs and CeHSFs were highly conserved, except CeHSF22, which had a 27-amino acid insertion segment in its α3-helix. Motif analysis of *C. ensifolium* HSFs showed that motif1, motif2, and motif4 were simultaneously present in most members, representing the major motifs constituting the *HSF* family (Figure 2B). Meanwhile, Groups B and C exhibited high similarity in conserved motifs, implying analogous functions for members of these two groups. However, different groups also had distinct conserved motifs, which might be related to their regulatory roles. Notably, Group C members only had 4 motifs (motifs 1, 2, 3, and 4), while Group B proteins contained motifs 1, 2, and 4, as well as motifs 5, 15, and 11. Motif 11 only was unique to Group B. Aside from the shared motifs, Group A also possessed motifs 6, 7, 8, 9, 10, 12, 13, and 14.

Exon–intron analysis uncovered exon numbers ranging from two to five, among the 22 *CeHSF* genes (Figure 2C). *CeHSF6* had the maximal five exons, while over half of the *CeHSF* genes (13, 59.1%) contained two exons. All *CeHSFs* exhibited one to four introns, with 13 genes harboring just one intron similar to the exon pattern. *CeHSF6* contained the most introns at four, implying greater functional diversity. Approximately 72.7% of the 22 *CeHSF* genes lacked UTRs, while *CeHSF18* had the most UTRs at three.

### 2.3. Secondary and Tertiary Structures of the CeHSF Proteins

The HSF protein’s secondary structure included an alpha helix, an extended strand, a beta turn, and a random coil. The secondary structure of the 22 CeHSF proteins was analyzed using SOPMA (Appendix A; Figure 3B). The results showed that alpha helices and random coils were the main secondary structural elements (alpha helix 34.27–56.63%, random coil 31.49–51.05%), followed by extended chains and folding (extended strand 7.18–10.14%, beta turn 4.55–6.85%). Similar results were found for the rest of the proteins (Appendix A).

The protein sequences were submitted to Alphafold2 online tool for tertiary structure prediction analysis (Appendix A). We selected three proteins with a certain degree of conservation to display the tertiary structures (Figure 3A). The secondary and tertiary structure results showed that CeHSF2, CeHSF3, and CeHSF22 had similar ratios of alpha helices, extended strands, beta turns, and random coils. The random coil ratio of CeHSF22 was markedly higher than that of other HSF proteins. The tertiary structure of the HSF protein was mainly composed of alpha helices, and the tertiary structures of CeHSF2, CeHSF3, and CeHSF22 were similar.

### 2.4. Chromosomal Distribution and Gene Duplication Events in CeHSFs

The number of *CeHSF* genes varied greatly across different chromosomes. The 22 *CeHSF* genes were distributed on 14 chromosomes, which were named based on their physical locations. Chr02 contained the most *CeHSF* genes (7, 31.8%), followed by Chr06 (3, 13.6%). The other 12 chromosomes each harbored only one *CeHSF* gene, while no *CeHSF* gene was found on Chr01, Chr08, Chr10, Chr15, Chr16, or Chr20 (Figure 4).

Gene duplication events, including tandem and segmental duplications, play vital roles in gene duplication and neofunctionalization. Analysis of duplication events in *CeHSFs* identified two tandem repeats on Chr02 and two segmental duplication pairs between Chr12 and Chr17 and between Chr12 and Chr19 (Figure 4). Integrated with phylogenetic analysis, genes involved in tandem and segmental duplications clustered together on the phylogenetic tree, implying expansion of these *CeHSFs* during evolution. Overall, these results indicated that most *CeHSFs* were relatively conserved, except for the duplication events on Chr02, Chr12, Chr17, and Chr19.

### 2.5. Protein–Protein Interaction Network Analysis of CeHSF Family Members

Protein–protein interaction (PPI) prediction was performed to gain further insight into the biological roles and regulatory networks of CeHSFs. A total of 51 proteins were detected, including 11 CeHSFs, that exhibited interactions, with 22 proteins interacting with CeHSFs (Figure 5A, Appendix A). Most proteins interacting with CeHSFs were functionally validated as related to heat stress, such as HSP90-1, HSP70-4, HSBP, and HSFA1A. Additionally, proteins involved in plant development or non-biotic stress responses, like MPK3, MPK6, ZAT6, FKBP62, and FKBP65, also interacted with CeHSFs. PPI network prediction further revealed potential interactions between different CeHSFs. Among the 11 CeHSFs, interactions were found among CeHSF2, CeHSF7, CeHSF10, CeHSF12, and CeHSF14 (Figure 5B), implying that CeHSFs may form complexes in response to stress.

### 2.6. Cis-Acting Elements in the Promoter Regions of CeHSF Genes

To understand the genetic functions and regulatory mechanisms of *CeHSF* genes, the cis-regulatory elements from *CeHSF* genes’ upstream promoter regions were predicted through the PlantPAN website. Thirty-six types of binding sites of transcription factor families were identified within 2000 bp upstream of the promoter in 22 *CeHSF* genes. These 36 categories appeared 12,297 times in the promoter regions of 22 *CeHSF* genes, among which AP2/ERF, AT-Hook, and GATA elements were the most prevalent (Figure 6A). The core promoter region of most *CeHSF* genes contained elements such as AP2/ERF, bHLH, bZIP, Dof, GATA, MYB, NF-YB/NF-YA/NF-YC, and ZF-HD. In addition, we also divided the promoter region into three sub-regions (1–500 kb, 501–100 kb, and 1001–2000 kb) (Figure 6B–D). We observed that AP2/ERF appeared 266 times in *CeHSF18*, accounting for 45.24% of the total number of loci in the gene. It was also the most abundant region among the 22 *CeHSF* genes, indicating that this gene might have been involved in plant growth, development, and stress responses. In addition, it is noteworthy that the LFY was only found in *CeHSF12* and the LOB/LBD was only found in *CeHSF11*, which means they might have been involved in cell proliferation, floral organ development, and nitrogen metabolism regulation. Overall, most genes had at least 20 different binding sites in the core region of the promoter, which were implicated in plant growth metabolism (AT Hook, AP2/ERF) and abiotic stress (bZIP, Dehydrin).

### 2.7. Excavating miRNA Targets for CeHSF Genes

We selected 22 members of the *CeHSF* gene family as candidate target gene sequences to predict their miRNAs. The results (Figure 7; Appendix A) showed that except for the *CeHSF11*, *CeHSF14*, and *CeHSF19* members of *CeHSFs*, other *CeHSF* family genes were predicted targets of at least one miRNA. *CeHSF18* was one of the most targeted *HSF* genes, predicted to be targeted by seven miRNAs. Most genes had at least one or two targeting binding sites of miRNAs (Figure 7). *CeHSF2*, *3*, and *6* were predicted to contain miR156 binding sites, a temperature-sensitive miRNA that coordinates plant flowering in response to temperature. Additionally, several miRNA binding sites related to plant defense mechanisms were identified, including sites for miR393 and miR319. These findings implicated the existence of a complex network regulation system between miRNA and *CeHSFs*.

### 2.8. Expression Profiles of HSF Genes in C. ensifolium Tissues and Development Stages

Based on transcriptomic data, the expression patterns of 22 *HSF* genes in *C. ensifolium* are shown in Figure 8. Among all 22 *CeHSF* genes, the expression levels of *CeHSF1*, *CeHSF2*, *CeHSF3, CeHSF4*, *CeHSF5*, *CeHSF6*, *CeHSF10*, and *CeHSF22* genes were very low in all samples, below detection limits. However, no gene showed significantly high expression in all samples. The *HSF* gene family exhibited certain tissue specificity in both vegetative and floral organs of *C. ensifolium* (Figure 8A,B). Among them, *CeHSF13* and *CeHSF15* showed relatively significant expression in flowers compared to other genes. Additionally, these two genes exhibited the highest expression levels in the gynandrium. The expression of *CeHSF11*, *CeHSF13*, *CeHSF15*, *CeHSF18*, and *CeHSF21* in vegetative organs was relatively higher. Specifically, *CeHSF13* displayed remarkably significant expression in roots, *CeHSF15* exhibited the most prominent expression in leaves, and *CeHSF21* showed the most significant expression in pseudobulbs. It is noteworthy that *HSF* genes exhibited certain expression patterns during the growth and development of flowers. For instance, in the process of floral bud development, the expression levels of *CeHSF7*, *CeHSF14*, *CeHSF18*, and *CeHSF21* showed an increasing trend (Figure 8C). During different developmental stages of flowers, the expression level of *CeHSF12* gradually increased, reaching its highest level during the peak flowering stage, and then gradually decreased as the flower deteriorated. *CeHSF15* also showed a similar trend of initially increasing and then decreasing expression, with the most significant expression occurring during the early opening stage (Figure 8D). Meanwhile, *CeHSF13* exhibited a trend of initially decreasing and then increasing expression. Its expression level decreased as the flower bud matured, and then gradually increased as the flower opened (Figure 8D). It is worth noting that *HSF* genes presented certain expression patterns during the growth and development of flowers. For instance, during the process of floral bud development, the expression levels of *CeHSF7*, *CeHSF14*, *CeHSF18*, and *CeHSF21* showed an increasing trend (Figure 8C). Throughout different developmental stages of flowers, the expression level of *CeHSF12* gradually increased, reaching its highest level during the peak flowering stage, and then gradually decreased as the flower deteriorated. *CeHSF15* also exhibited a similar trend of initially increasing and then decreasing expression, with the most significant expression occurring during the initial opening stage (Figure 8D). Meanwhile, *CeHSF13* displayed a trend of initially decreasing and then increasing expression. Its expression level decreased as the flower bud matured, and then gradually increased as the flower opened (Figure 8D).

### 2.9. Expression of CeHSFs in Response to Different Temperature Treatments

Seven genes with relatively higher expression in roots, pseudobulbs, leaves, bracts, and different flower development stages and other parts were selected as target genes. These genes were most likely to respond to the regulation of expression under temperature stress (Figure 9). The results of high-temperature treatment showed that the expression of *CeHSF18* was sharply induced and remained highly expressed. *CeHSF7*, *CeHSF14*, *CeHSF15,* and *CeHSF21* exhibited a consistent pattern, with significant up-regulation after 6 h of high-temperature treatment, rapid down-regulation after 12 h, and elevated expression again after 24 h. Interestingly, *CeHSF14* showed a nearly 200-fold up-regulation after 6 h of high-temperature treatment (Figure 9). At 12 h of high-temperature treatment, the expression of *CeHSF11* and *CeHSF13* was continuously inhibited. But after 24 h, their expression was higher than that before high-temperature treatment. The results of low-temperature treatment showed a similar pattern for *CeHSF7*, *CeHSF11*, *CeHSF13*, *CeHSF14,* and *CeHSF18*. *CeHSF7*, *CeHSF11,* and *CeHSF18* were highly expressed at 12 h of low temperature, with a decrease in expression after 24 h. *CeHSF13* and *CeHSF14* showed the highest expression at 6 h of low temperature, followed by down-regulation. In the low-temperature environment, the expression of *CeHSF15* and *CeHSF21* continuously decreased. Gene expression was significantly inhibited. Interestingly, *CeHSF15* exhibited repaid induction and stepwise suppression during cold temprature, this inferred opposite models of regulation. Combining the transcriptome data, *CeHSF15* expressed earlier than the presence of floral scents. In the process of flowering, *CeHSF15* may act as a trigger in the lanch of flower scents. 

Further correlation analysis indicates a highly significant correlation between *CeHSF14*, *CeHSF7*, *CeHSF15,* and *CeHSF21*. The expression of these four genes may have a certain relationship. *CeHSF11* was significantly correlated with *CeHSF13*, suggesting a potential relationship in the expression of these two genes (Figure 9).

## 3. Discussion

### 3.1. Conservation and Expansion of CeHSF Family Members

*HSF* genes play a crucial role in various aspects of plant growth, development, and response to different stresses, including salt, heat, and cold stress [39,40]. Therefore, diversity in *HSF* family members has long been a subject of considerable attention. Currently, *HSF* genes have been identified in numerous plant species, including *Brassica napus* (64 members) [41], *Populus trichocarpa* (28 members), and *Medicago truncatula* (16 members) [42]. The number of members in the *HSF* family varies in different *Gossypium* species, ranging from 31 to 78 [43]. Similarly, monocotyledonous plants such as *Triticum aestivum*, rye, rice, and maize have 82, 31, 22, and 25 identified members [32,33,34,44], respectively. Discrepancies in the number of *HSF* family members among different plants may be attributed to the differential retention of *HSF* genes during the evolutionary process to adapt to the environment. In this study, 22 *HSF* gene family members were identified in *C. ensifolium*, a number similar to some monocot plants such as rice and maize [32,34] (Figure 1). CeHSFs displayed a high degree of conservation in their sequences (Figure 2A). However, some duplication was observed in the A2 and B4 subfamilies (Figure 1, Figure 4). The diversity of HSF family members among different plants might be linked to whole-genome duplication (WGD) events in plants, as seen in the differences between members in *A. thaliana* and soybean [45,46]. *C. ensifolium* has undergone WGD events in its evolutionary history [47], which might explain the expansion of *CeHSF* family members.

### 3.2. Functional Prediction and Abiotic Stress Response of CeHSFs

Extreme environments currently threaten plant survival and development worldwide. MicroRNAs (miRNAs) are short non-coding RNAs that have emerged as key post-transcriptional regulators in many species. Under extreme conditions, miRNAs are one of the mechanisms for plant stress responses [48,49]. Previous studies have shown miRNA involvement in abiotic stresses including drought, salinity, and temperature [48,50,51]. Our investigation revealed that, with the exception of *CeHSF11*, *14*, and *19*, every other member within the *CeHSF* family harbored a minimum of one miRNA binding site (Appendix A). *CeHSF2*, *3*, and *6* were predicted to contain miR156, a temperature-sensitive miRNA that coordinates plant flowering in response to temperature [51]. Additionally, many miRNA binding sites associated with plant defense mechanisms were present, such as miR393 and miR319 [52]. These results indicate the important roles of *CeHSFs* in responding to stresses, especially temperature stresses. Similar to previous studies [33,53], protein–protein interaction network analysis further revealed interactions of CeHSFs with heat stress-related proteins such as HSP70-4.

Plants can respond to environmental stresses and regulate growth via tissue-specific gene expression [54,55]. Previous reports indicated differential expression of *HSF* genes in some species [56,57]. In *C. ensifolium*, *HSF* genes showed specific expression patterns in different tissues (Figure 8). For instance, *CeHSF13* was significantly expressed in roots and bracts, while *CeHSF15* exhibited higher expression in leaves. Both were specifically expressed in flowers. Notably, during the growth of floral buds, the expression of *CeHSF13* gradually decreased, whereas *CeHSF15* showed an initial increase followed by a decrease during the transition from floral bud to open flower. Moreover, several other *CeHSF* genes showed significant expression, indicating that *CeHSFs* could have important roles in the growth and development of multiple organs and tissues in *C. ensifolium*.

In previous studies, *HSFs* have been shown to play major roles in plants’ responses to abiotic stress, enhancing their thermotolerance and salt stress tolerance [16,58,59]. In *Dianthus caryophyllus*, the majority of *DcaHSFs* were responsive to heat stress, while some genes were downregulated by cold stress [60]. *HSF* genes in *Hypericum perforatum* showed pronounced up-regulation under heat stress [61]. In *Zingiber officinale*, *ZoHSFs* exhibited an expression pattern of initial upregulation followed by down-regulation under high temperature and strong light stress [62]. In this study, seven *CeHSF* genes were treated with high- and low-temperature treatments. The qRT-PCR results under high-temperature conditions showed the rapid up-regulation of *CeHSF18* expression, while *CeHSF14* exhibited significant up-regulation after 6 h of high-temperature treatment (Figure 9). Additionally, many genes showed varying degrees of expression, consistent with previous research [63], suggesting the involvement of *CeHSF* genes in the heat stress tolerance of *C. ensifolium*. Under low-temperature treatment, *CeHSF7*, *CeHSF11*, *CeHSF14*, *CeHSF13*, and *CeHSF18* all exhibited up-regulation, indicating their potential role in regulating the plant response to low-temperature stress (Figure 9). However, the expression of *CeHSF15* and *CeHSF21* continuously decreased during low-temperature stress, suggesting that *C. ensifolium* may mitigate cold damage by reducing the expression of heat stress transcription factors.

## 4. Materials and Methods

### 4.1. Identification and Physicochemical Properties of HSF Genes in C. ensifolium Genome

The genomic sequence and annotation data for *C. ensifolium* [47] were downloaded from the National Genome Data Center (NGDC) (https://ngdc.cncb.ac.cn/, accessed on 22 October 2023). The HSF protein sequences of *Arabidopsis* were downloaded from TAIR (http://www.arabidopsis.org/, accessed on 22 October 2023). The HSF protein sequences of *Arabidopsis* were used as query sequences to execute a BLASTP search against the *C. ensifolium* genome (E-value < 1 × 10^−5^, Num of Hits: 500, Num of Aligns: 250). The BLASTP results are shown in Appendix A. The Hidden Markov model (HMM) with PF00447 (HSF-type DBD domain) was used to match the *HSF* gene sequences in *C. ensifolium* through the TBtools (version 2.019) [64] software and domain checking was performed to remove sequences without DBD domains. ProtParam online analytical tools (https://web.expasy.org/protparam/, accessed on 23 October 2023) were used to predict the number of amino acids, molecular weight, theoretical pI, instability index, aliphatic index, and grand average of hydropathicity.

### 4.2. Analysis of HSF Protein Phylogenetic Relationships and Conserved Domains

The protein sequences for OsHSFs and PeHSFs were retrieved from the HSF (HEATSTER, http://www.cibiv.at/services/hsf/, accessed on 22 October 2023) database and Wang et al. [53]. A neighbor-joining (NJ) phylogenetic tree of *A. thaliana*, *O. sativa*, *Passiflora edulis,* and *C. ensifolium* was constructed using PhyloSuite (version 1.2.3) [65] with 1000 bootstrap replicates. The phylogenetic tree was visualized using the iTOL website (https://itol.embl.de/, accessed on 23 October 2023). The online software NCBI BatchCD-search (https://www.ncbi.nlm.nih.gov/Structure/bwrpsb/bwrpsb.cgi, accessed on 25 October 2023) was used to analyze the conserved domains. MEME (https://meme-suite.org/meme/doc/, accessed on 25 October 2023) was used to analyze the conserved motifs. Multiple sequence alignments were generated with PhyloSuite (version 1.2.3) and visualized by ESPript 3.0 [66].

### 4.3. Protein Interactions and Chromosome Distribution Analysis of the CeHSF Genes

Protein–protein interaction network prediction analysis was conducted using the STRING database and Cytoscape (version 3.10) [67]. The 22 CeHSF protein sequences were submitted to the STRING database (http://string-db.org/cgi, accessed on 28 October 2023) using *Arabidopsis* orthologs as references, with no more than 20 interactors (1st and 2nd shell). All *CeHSF* genes were mapped to locations on different *C. ensifolium* chromosomes using the TBtools software (version 2.019). The syntenic relationship between the genes and replication events was analyzed by the Run MCScanx Wrapper function in TBtools (version 2.019) and visualized by Excel2021 and TBtools software (version 2.019).

### 4.4. Promoter Capture and Prediction of Specific miRNA targets in the CeHSF Family Members

TBtools (version 2.019) was used to obtain a 2000 bp sequence upstream of the *HSF* genes in *C. ensifolium* from the start codon. Cis-acting elements in the promoter region of the *CeHSF* were analyzed using PlantPAN4.0 (http://plantpan.itps.ncku.edu.tw/plantpan4, accessed on 29 October 2023) [68]. The data were analyzed and visualized by Excel2021. Bioinformatics and prediction analyses of miRNAs and their target *CeHSF* genes were performed in the web-based psRNA Target Server (https://www.zhaolab.org/psRNATarget/analysis, accessed on 29 October 2023). The expected value was set to 4.5 and the remaining parameters were set to default. Finally, alignment of identified genes with the miRNAs of *A. thaliana* was conducted.

### 4.5. Prediction of Secondary and Tertiary Structures of the CeHSF Transcription Factor Proteins

In this study, we used online tools, SOPMA (https://npsaprabi.ibcp.fr, accessed on 30 October 2023), and the CFSSP database (https://www.biogem.org/tool/chou-fasman/, accessed on 30 October 2023) for secondary structure [69] and Alphafold2 (https://colab.research.google.com/github/deepmind/alphafold/blob/main/notebooks/AlphaFold.ipynb, accessed on 30 October 2023) to predict the tertiary structures of the 22 CeHSF members. The tertiary structures of proteins were visualized by PyMOL (Version 2.5.7).

### 4.6. Analysis of Gene Expression Patterns

In order to investigate the potential involvement of *HSF* genes in various organs of *C. ensifolium*, we downloaded the RNA-seq data of *C. ensifolium* from the National Genomics Data Center. The FPKM values of different *CeHSF* genes and the Genome Sequence Archive (GSA) RUN accession numbers were presented in Appendix A. Subsequently, RNA-Seq reads were mapped to the *C. ensifolium* CDS files, and gene expression was calculated using kallisto (version 0.48.0) with default parameters [70,71]. The gene expression patterns were visualized using TBtools (version 2.019).

### 4.7. The Assay of qRT-PCR during the High- and Low-Temperature Induction

The materials for this experiment were obtained from the cultivation of the *C. ensifolium* variety ‘Xiao Tao Hong’ at the germplasm resource nursery of Fujian Agriculture and Forestry University (26°04′51.3″ N, 119°14′19.9″ E). To ensure the consistency of the experiment and the stability and repeatability of the results, independent plants were selected from the nursery. Four individuals were prepared as biological replicates for each treatment. Untreated plants with similar growth status were used as controls. Plants were subjected to stress by high and low temperatures, and tender leaves from the plant tops were collected at corresponding time points. The treatment involved high-temperature stress at 42 °C and low-temperature stress at 4 °C in incubators. Tender leaves of the orchid ‘Xiao Tao Hong’ were sampled after 6, 12, and 24 h of high- and low-temperature treatments. The samples were placed in 1.5 mL sterile non-enzymatic cryopreservation tubes and rapidly frozen in liquid nitrogen. Finally, total RNA was extracted by using the R6827 Plant RNA Kit (Omega Bio-Tek, Guangzhou, China). DNA digestion was performed to remove DNA from the total RNA extracts. The Hieff UNICON Universal Blue qPCR SYBR Green Master Mix kit (Yeasen Biotechnology, Shanghai, China) was used for reverse transcription to synthesize quantitative cDNA single strands from 2 mg RNA. This cDNA served as the template for real-time quantitative PCR detection. The *CeTUB* gene was used as the internal reference using fluorescence quantification. The gene sequences and internal reference primers used in the reaction are shown in Appendix A. The reaction was designed with three technical replicates. The reaction system totaled 20 μL, comprising 10 μL of Hieff UNICON Universal Blue qPCR SYBR Green Master Mix, 0.4 μL of forward primer (10 μM), 0.4 μL of reverse primer (10 μM), 4 μL of template DNA, and 5.2 μL of sterile ultrapure water. The amplification program consisted of a pre-denaturation at 95 °C for 2 min, followed by 40 cycles of denaturation at 95 °C for 10 s and annealing/extension at 60 °C for 30 s. Data were calculated using the 2^−ΔΔCt^ method to determine the relative gene expression levels. A one-way analysis of variance (ANOVA) was applied for data processing. Statistical differences were compared using *t*-tests based on IBM SPSS Statistics 24, with *p* < 0.05 as * and *p* < 0.01 as **. Finally, visual analysis was performed using Origin and Chiplot (https://www.chiplot.online/, accessed on 15 November 2023).

## 5. Conclusions

In this study, 22 *CeHSF*s were classified into three groups. Tandem and segmental duplications among *CeHSFs* represented the major expansion of this gene family. Examination of protein interaction networks, promoter cis-acting elements, and miRNA-splicing sites provided insights into the intricate stress response regulation mediated by *CeHSFs*. Expression profiling across tissues suggested potential regulatory roles of *CeHSFs* in orchid development. High and low temperature stress experiments further demonstrated the significance of *CeHSFs* in modulating *C. ensifolium*’s responses to various abiotic stimuli. *CeHSF14* was shown to be the candidate TF responding to high- and low-temperature stress through possible miRNA control. 

These results further underscore the significant role of *HSF* genes in the plant response to abiotic stress. Given the vast consumer market for *C. ensifolium*, breeding new varieties with strong stress resistance is of paramount importance. Therefore, studying *CeHSFs* will contribute to advancing breeding efforts in this regard.

## Figures and Tables

**Figure 1 ijms-25-01002-f001:**
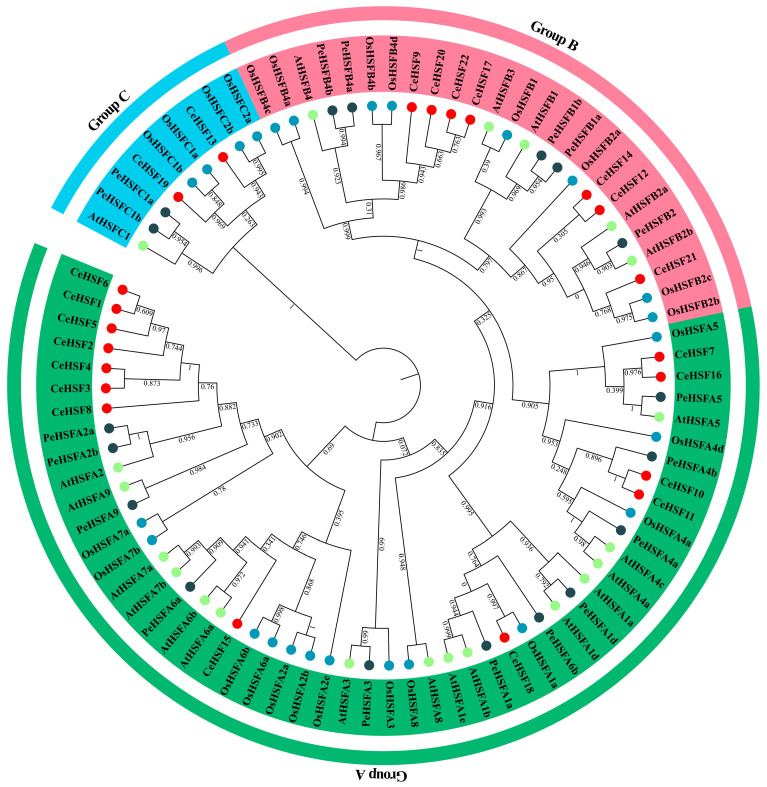
Phylogenetic analysis of *HSF* genes in *Cymbidium ensifolium*. The neighbor-joining (NJ) phylogenetic tree was constructed using PhyloSuite with 1000 bootstraps. Group A, Group B, and Group C were grouped together as indicated in green, red, and blue, respectively. Different colored circles represent different species; red: *C. ensifolium*, green: *A. thaliana*, blue: *O. sativa*, and dark blue: *Passiflora edulis*.

**Figure 2 ijms-25-01002-f002:**
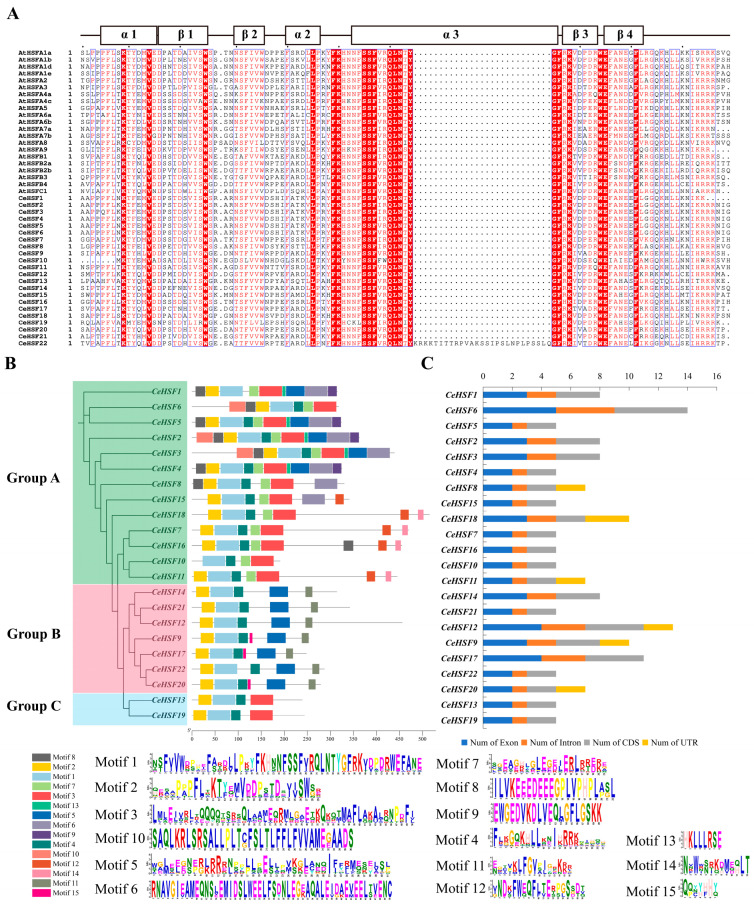
Multiple sequence alignment, gene structures, and conserved motif analysis in *CeHSFs*. (**A**) Multiple sequence alignment of the DBD domains among *AtHSFs* and *CeHSFs*. Red box and white character represent high homology. Red character or black bold character represent similarity within a group (column). Blue frame represents similarity across groups (columns). (**B**) Phylogenetic relationships and motif compositions of *Cymbidium ensifolium* HSF proteins. (**C**) Statistical analysis of introns, exons, CDSs, and UTRs in *CeHSFs***.**

**Figure 3 ijms-25-01002-f003:**
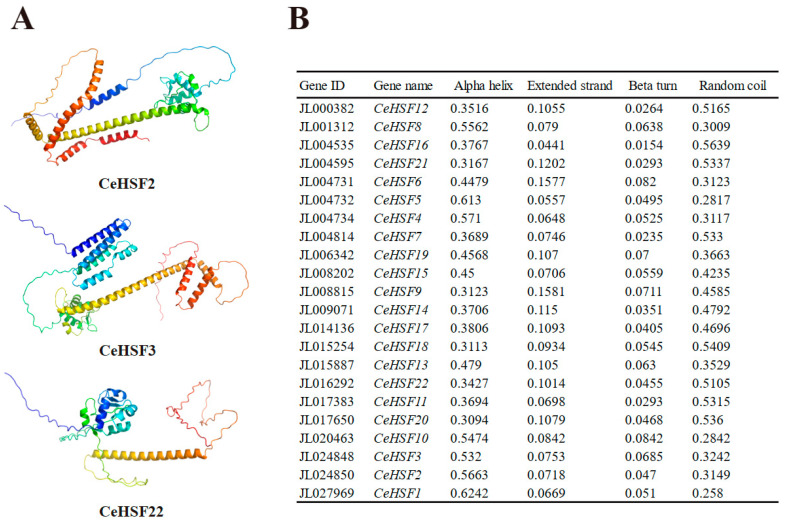
The prediction protein structures of CeHSF proteins. (**A**) Tertiary structure of 3 CeHSF proteins. (**B**) Secondary structure ratio of 22 CeHSF proteins. The complete secondary and tertiary structures of 22 CeHSF proteins are displayed in Appendix A.

**Figure 4 ijms-25-01002-f004:**
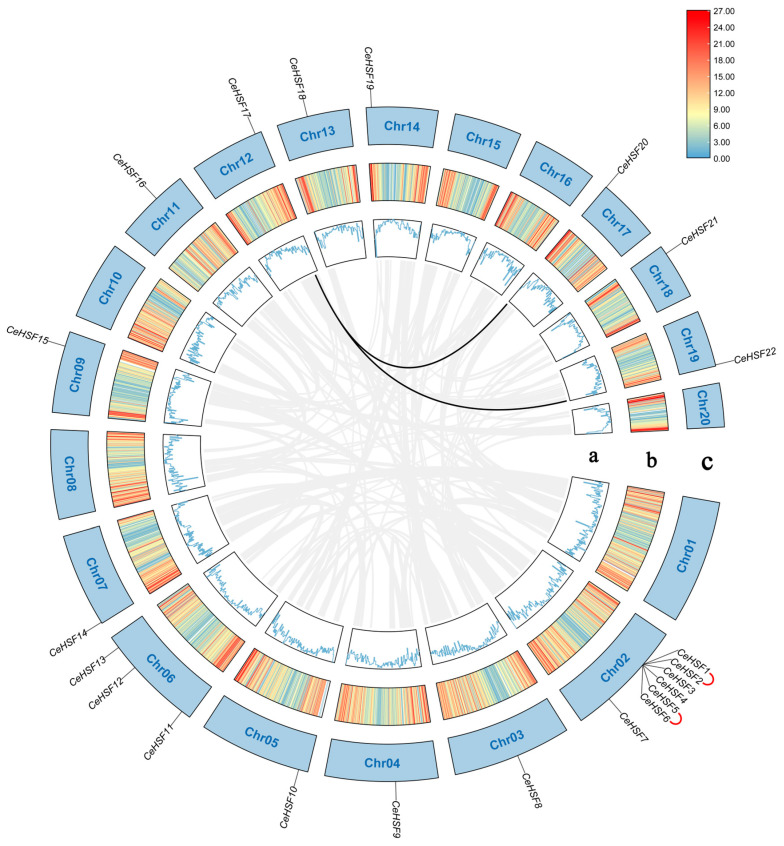
Chromosome localization and synteny analysis of *CeHSF* genes. Genes with tandem repeats were linked externally with red lines, and genes with segment repeats were linked internally with black lines. a: The blue line represents gene density. b: Line colors indicate gene density. c: Chr01–Chr20 represents the twenty chromosomes of *Cymbidium ensifolium*.

**Figure 5 ijms-25-01002-f005:**
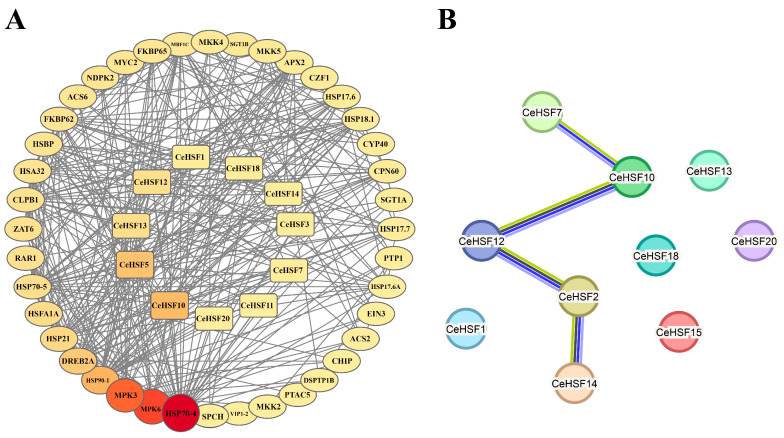
Predicted protein–protein interaction networks of CeHSF proteins. (**A**) Protein interaction networks of CeHSF proteins with other proteins. The outer circle represents proteins that interact with CeHSFs. The inner circle represents CeHSF proteins. The interaction between these proteins was represented by the gray lines. Darker colors corresponded to stronger interactions between proteins. (**B**) Protein interaction networks of only CeHSF proteins.

**Figure 6 ijms-25-01002-f006:**
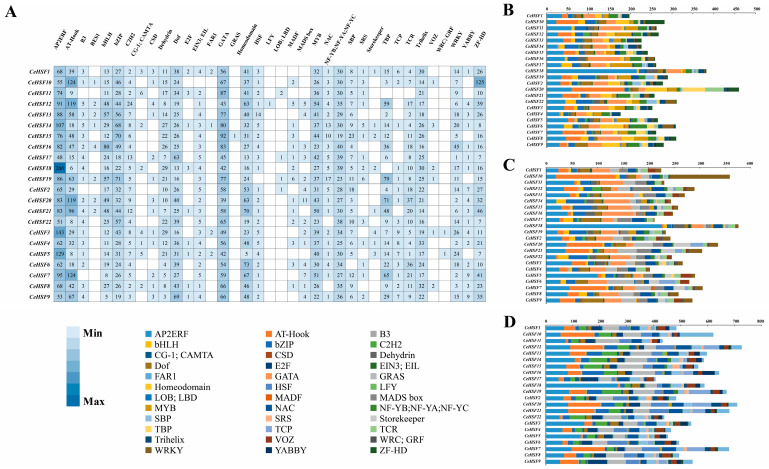
Analysis of cis-acting elements in *CeHSF* promoters. (**A**) Heatmap analysis of cis-acting elements of 2 kb promoter regions of CeHsf genes. (**B**) Cis-acting elements with starting positions from 1 to 500 kb. (**C**) Cis-acting elements with starting positions from 501 to 1000 kb. (**D**) Cis-acting elements with starting positions from 1001 to 2000 kb.

**Figure 7 ijms-25-01002-f007:**
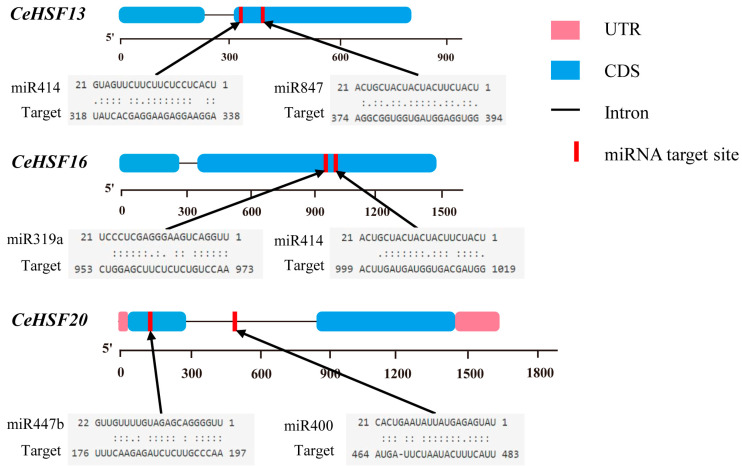
Prediction of targets for some microRNAs. This view displays partial regulatory roles of miRNA and CeHSF transcription factors. Blue parts show the coding region of *CeHSFs*. Red lines mark the splicing sites. The complete results are displayed in Appendix A.

**Figure 8 ijms-25-01002-f008:**
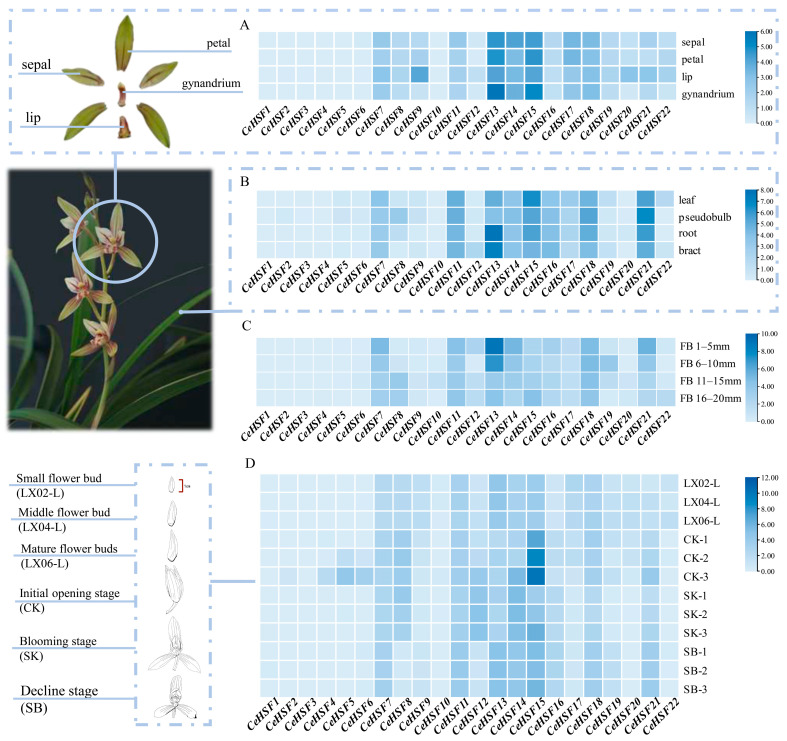
Heatmap of the expression patterns of the *HSF* gene family in *Cymbidium ensifolium*. (**A**) Heatmap of floral organs in *Cymbidium ensifolium*. (**B**) Heatmap of vegetative organs in *Cymbidium ensifolium*. (**C**) Heatmap of flower buds at different stages. FB 1–5 mm: flower bud of 1–5 mm, FB 6–10 mm: flower bud of 6–10 mm, FB 11–15 mm: flower bud of 11–15 mm, FB 16–20 mm: flower bud of 16–20 mm. (**D**) Heatmap of different flower development stages. Small flower bud: LX02-L, Middle flower bud: LX04-L, Mature flower buds: LX06-L, Initial opening stage: CK, Blooming stage: SK, Decline stage: SB.

**Figure 9 ijms-25-01002-f009:**
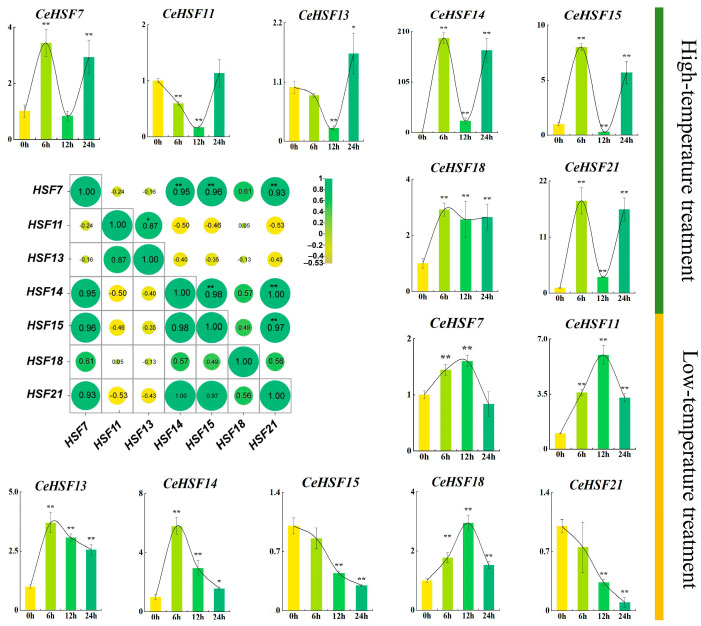
Expression analysis of the 7 *CeHSF* genes in leaves under different abiotic stresses (heat and cold treatments). Heatmap in the middle shows the correlation of gene expressions between the two treatments. Data are means ± SE of three separate measurements based on *t*-test, taking *p* < 0.05 as * and *p* < 0.01 as **.

**Table 1 ijms-25-01002-t001:** Protein information of *HSF* gene family in *Cymbidium ensifolium*, including gene ID, gene name, genomics position, and protein aphysicochemical properties.

Gene ID	Gene Name	Chr NO.	Location	Number of Amino Acid	Molecular Weight (kDa)	Theoretical pI	Instability Index	Aliphatic Index	Grand Average of Hydropathicity
JL027969	*CeHSF1*	Chr02	70,865,430–70,866,480	314	35.76	4.87	54.65	83.25	−0.493
JL020463	*CeHSF10*	Chr05	17,677,243–17,678,347	190	22.40	6.66	39.08	88.74	−0.556
JL017383	*CeHSF11*	Chr06	18,140,474–18,142,900	444	50.07	5.33	52.71	70.86	−0.612
JL000382	*CeHSF12*	Chr06	138,769,982–138,773,656	455	50.16	5.88	72.56	71.05	−0.506
JL015887	*CeHSF13*	Chr06	187,008,263–187,009,061	238	27.06	8.5	50.42	68.03	−0.561
JL009071	*CeHSF14*	Chr07	2,167,682–2,170,116	313	35.13	5.15	52.03	71.25	−0.554
JL008202	*CeHSF15*	Chr09	126,380,829–126,391,385	340	39.25	5.1	53.85	68.88	−0.899
JL004535	*CeHSF16*	Chr11	85,935,326–85,936,780	454	51.52	4.91	56.42	70.24	−0.785
JL014136	*CeHSF17*	Chr12	131,441,695–131,443,308	247	28.96	9.26	57.07	74.94	−0.664
JL015254	*CeHSF18*	Chr13	58,995,544–59,061,011	514	56.61	4.82	55.52	73.21	−0.5
JL006342	*CeHSF19*	Chr14	10,028,132–10,028,944	243	28.42	6.93	72.46	79.79	−0.682
JL024850	*CeHSF2*	Chr02	70,966,809–70,972,457	362	41.52	5.44	59.52	84.34	−0.472
JL017650	*CeHSF20*	Chr17	619,190–620,825	278	32.30	6.39	48.08	68.35	−0.671
JL004595	*CeHSF21*	Chr18	16,224,229–16,225,346	341	37.88	4.88	64.83	68.09	−0.56
JL016292	*CeHSF22*	Chr19	105,363,634–105,365,001	286	32.83	9.44	57.32	76.29	−0.567
JL024848	*CeHSF3*	Chr02	71,041,003–71,044,452	438	50.05	4.99	67.57	77.28	−0.414
JL004734	*CeHSF4*	Chr02	71,806,634–71,807,686	324	37.17	4.85	64.35	79.48	−0.611
JL004732	*CeHSF5*	Chr02	71,910,724–71,911,773	323	36.71	4.95	59.47	82.14	−0.523
JL004731	*CeHSF6*	Chr02	71,912,381–71,987,374	317	36.22	9.61	56.96	85.52	−0.396
JL004814	*CeHSF7*	Chr02	161,729,330–161,730,829	469	52.48	5.03	57.83	66.74	−0.688
JL001312	*CeHSF8*	Chr03	103,648,460–103,772,894	329	37.29	5.41	39.73	72.04	−0.574
JL008815	*CeHSF9*	Chr04	80,548,866–80,553,632	253	29.46	6.56	55.48	82.81	−0.409

## Data Availability

The raw genome data and assembled *C. ensifolium* genome were submitted to the National Genomics Data Center (NGDC) database with the accession number PRJCA005355/CRA004327 and GWHBCII00000000. The raw transcriptome sequences have been deposited in the BioProject of GSA under the accession codes PRJCA009885/CRA007101 and PRJCA005426/CRA004351, respectively. All data generated or analyzed during this study are included in this published article (Appendix A) and also available from the corresponding author upon reasonable request.

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
