# Peer review of "General Analysis of Heat Shock Factors in the Cymbidium ensifolium Genome Provided Insights into Their Evolution and Special Roles with Response to Temperature"

_ijms, 2024, doi:10.3390/ijms25021002_

Round 1
Reviewer 1 Report
Comments and Suggestions for Authors
The authors systematically analyzed the CeHSF genes of Cymbidium ensifolium, and found some of the genes are involved in tissue development and response to temperature. The story is interesting and there's significance for further studying the functions of these genes in C. ensifolium. But the manuscript need be improved in the follow areas:
1、 The CeHSF had been compared with other three species in evolution analysis. They had been divided into three classes, and furtherly been divided into 15 subclasses. There are some problems here. For e.g: (1) In A7, AtHSF and OsHSF can’t be clustered together, and the authors also mentioned there’s some deviation in their description and showing in Fig 1. (2) They described “A1, A6, C1 and C2 each had only 1 member”, but we can find both 2 members in C1 and C2 in Fig 1. In generally, the guidance value needs to be greater than 95% to be trusted. In Figure 1, the value of C2 is only 0.263, so the cluster need be modified. The classification in phylogenetic tree need be revised.
2、 Conservative domains are the concept of proteins. The description in Line359 is not accurate which mentioned the conserved domains of genes, which need be revised to the genes encoded proteins.
3、 Some experiments (In vitro / in vivo) are recommended to supplement. Such as subcellular localization and overexpression of some of these genes in Arabidopsis or tobacco.
Based on these I recommend a major revision. If the authors invite native English speakers to revise the language, it’ll be appreciated.
Comments on the Quality of English LanguageThe language need be improved!
Author Response
Comments and Suggestions for Authors
The authors systematically analyzed the CeHSF genes of Cymbidium ensifolium, and found some of the genes are involved in tissue development and response to temperature. The story is interesting and there's significance for further studying the functions of these genes in C. ensifolium. But the manuscript need be improved in the follow areas:
1、The CeHSF had been compared with other three species in evolution analysis. They had been divided into three classes, and furtherly been divided into 15 subclasses. There are some problems here. For e.g: (1) In A7, AtHSF and OsHSF can’t be clustered together, and the authors also mentioned there’s some deviation in their description and showing in Fig 1. (2) They described “A1, A6, C1 and C2 each had only 1 member”, but we can find both 2 members in C1 and C2 in Fig 1. In generally, the guidance value needs to be greater than 95% to be trusted. In Figure 1, the value of C2 is only 0.263, so the cluster need be modified. The classification in phylogenetic tree need be revised.
R: Thank you for your review and valuable suggestions. Your suggestions made a lot of sense, so we revised the Figure 1. Due to the differences in HSFs between monocotyledonous and dicotyledonous plants, certain genes did not cluster together. We concurred with your perspective and, upon reviewing more References [1-3], reclassified CeHSFs. The new classification is depicted in Figure 1. Furthermore, we rectified inaccuracies in the descriptions associated with Figure 1.
Line 86. we revised it as “Phylogenetic analysis showed that the 22 CeHSF members were divided into 3 groups (Group A, Group B and Group C)(Figure 1). This phylogenetic classification was validated by the presence of conserved DNA-binding domains (DBDs) and the established classifi-cation of Arabidopsis HSFs. Group A had the most C. ensifolium HSF members with 13, while Group C had the least with only 2. There were 7 C. ensifolium HSF members in Group B. The phylogenetic tree results highlighted some differences between monocot and eudicot HSFs, such as eudicots having 1 clade in Group C (C1), while monocots had two (C1 and C2). ”
[1] Guo Q, Wei R, Xu M, Yao W, Jiang J, Ma X, Qu G, Jiang T. Genome-wide analysis of HSF family and overexpression of PsnHSF21 confers salt tolerance in Populus simonii × nigra. Front Plant Sci. 2023 Apr 26;14:1160102. doi: 10.3389/fpls.2023.1160102. PMID: 37200984; PMCID: PMC10187788.
[2] Wang L, Liu Y, Chai G, Zhang D, Fang Y, Deng K, Aslam M, Niu X, Zhang W, Qin Y, Wang X. Identification of passion fruit HSF gene family and the functional analysis of PeHSF-C1a in response to heat and osmotic stress. Plant Physiol Biochem. 2023 Jul;200:107800. doi: 10.1016/j.plaphy.2023.107800. Epub 2023 May 25. PMID: 37253279.
[3] Ren, Y., Ma, R., Xie, M. et al. Genome-wide identification, phylogenetic and expression pattern analysis of HSF family genes in the Rye (Secale cereale L.). BMC Plant Biol 23, 441 (2023). https://doi.org/10.1186/s12870-023-04418-1
2、Conservative domains are the concept of proteins. The description in Line359 is not accurate which mentioned the conserved domains of genes, which need be revised to the genes encoded proteins.
R: Thank you for your review and valuable suggestions.
Line 359. we revised it as “Analysis of HSF protein phylogenetic relationships and conserved domains”.
3、Some experiments (In vitro / in vivo) are recommended to supplement. Such as subcellular localization and overexpression of some of these genes in Arabidopsis or tobacco.
R: Thank you for your review and valuable suggestions. Your suggestion was very meaningful. Due to some issues with laboratory equipment and the expression vectors required for the experiments, it was not possible to conduct the aforementioned experiments at present. However, these experiments were part of our next steps, and we plan to carry them out in the new article.
Based on these I recommend a major revision. If the authors invite native English speakers to revise the language, it’ll be appreciated.
R: Thank you for your review and valuable suggestions. Your suggestions made a lot of sense, so we revised the entire manuscript. We corrected some mistakes in words and grammar. We hope that this revision concurs your expectations.
Reviewer 2 Report
Comments and Suggestions for Authors
The manuscript entitled “General analysis of CeHSF genes in the Cymbidium ensifolium genome provided insights into their evolution and specific roles in tissue development and response to temperature” identify CeHSF genes from previously published C. ensifolium genome and analyze their phylogeny, gene structure, motif composition, expression patterns, and cis-elements. In total the manuscript needs minor revision to be ready for further consideration. With comments:
1- The abstract effectively outlines the identification and roles of CeHSF genes in Cymbidium ensifolium. However, the absence of quantitative data limits its clarity and immediate impact. Integrating key quantitative findings into the abstract would enhance its overall effectiveness.
2- While the manuscript offers valuable insights into Cymbidium ensifolium's heat shock factors (HSFs), there is a notable absence of background information on the species' biology and specific challenges, especially in hot regions and the problem of heat stress . Incorporating this context would enhance the manuscript's significance and make it more accessible to a broader readership.
3- It is crucial to acknowledge and cite several closely related works that appear relevant to the current study but are currently absent from the references. Incorporating these works would strengthen the manuscript by placing the findings within a broader context and acknowledging existing contributions to the field:
Li, X., Liu, L., Sun, S., Li, Y., Jia, L., Ye, S., ... & Luan, Y. (2023). Transcriptome analysis reveals the key pathways and candidate genes involved in salt stress responses in Cymbidium ensifolium leaves. BMC Plant Biology, 23(1), 64.
Wang, M. J., Ou, Y., Li, Z., Zheng, Q. D., Ke, Y. J., Lai, H. P., ... & Ai, Y. (2023). Genome-Wide Identification and Analysis of bHLH Transcription Factors Related to Anthocyanin Biosynthesis in Cymbidium ensifolium. International Journal of Molecular Sciences, 24(4), 3825.
4- The origin of the analyzed sequences is not explicitly stated in the manuscript. It would be beneficial to clarify whether the authors obtained these sequences through their own sequencing efforts or utilized a previously published genome. If the latter, please provide the relevant citation for the used published reference. Including this information in the Materials and Methods section would enhance the transparency of the study's methodology.
5- In the Materials and Methods section, it is essential to include comprehensive details regarding the utilized information, including default parameters of employed software. This should encompass specifics such as Gene IDs of the sequences, especially for reference genes in expression analysis, gap penalty, gap extension values, and the exact version of software. If online tools were used, please provide the access date. Providing this detailed information will enhance the reproducibility and transparency of the study's methodology
6- The arrowheads in Figure 7 appear to be improperly included, leading to potential confusion. I recommend a careful revision of the figure to ensure accurate representation.
7- The Neighbor-Joining phylogenetic tree would benefit from the inclusion of HSP genes from other orchid species. Integrating a broader range of orchid HSP sequences could provide a more comprehensive perspective on the evolutionary relationships within this gene family. Considering the relevance of comparative analysis, including additional orchid species in the NJ phylogenetic tree would strengthen the study's insights into the evolutionary dynamics of HSP genes in orchids.
9- The manuscript needs moderate language polishing which could be made as a final check.
Despite the aforementioned comments, I rate the manuscript as strong and strongly recommend its publication after minor revisions to address the raised questions. The proposed revisions would further enhance the clarity and completeness of the study, contributing to its overall strength and scientific meriف
Comments on the Quality of English LanguageThe manuscript entitled “General analysis of CeHSF genes in the Cymbidium ensifolium genome provided insights into their evolution and specific roles in tissue development and response to temperature” identify CeHSF genes from previously published C. ensifolium genome and analyze their phylogeny, gene structure, motif composition, expression patterns, and cis-elements. In total the manuscript needs minor revision to be ready for further consideration. With comments:
1- The abstract effectively outlines the identification and roles of CeHSF genes in Cymbidium ensifolium. However, the absence of quantitative data limits its clarity and immediate impact. Integrating key quantitative findings into the abstract would enhance its overall effectiveness.
2- While the manuscript offers valuable insights into Cymbidium ensifolium's heat shock factors (HSFs), there is a notable absence of background information on the species' biology and specific challenges, especially in hot regions and the problem of heat stress . Incorporating this context would enhance the manuscript's significance and make it more accessible to a broader readership.
3- It is crucial to acknowledge and cite several closely related works that appear relevant to the current study but are currently absent from the references. Incorporating these works would strengthen the manuscript by placing the findings within a broader context and acknowledging existing contributions to the field:
Li, X., Liu, L., Sun, S., Li, Y., Jia, L., Ye, S., ... & Luan, Y. (2023). Transcriptome analysis reveals the key pathways and candidate genes involved in salt stress responses in Cymbidium ensifolium leaves. BMC Plant Biology, 23(1), 64.
Wang, M. J., Ou, Y., Li, Z., Zheng, Q. D., Ke, Y. J., Lai, H. P., ... & Ai, Y. (2023). Genome-Wide Identification and Analysis of bHLH Transcription Factors Related to Anthocyanin Biosynthesis in Cymbidium ensifolium. International Journal of Molecular Sciences, 24(4), 3825.
4- The origin of the analyzed sequences is not explicitly stated in the manuscript. It would be beneficial to clarify whether the authors obtained these sequences through their own sequencing efforts or utilized a previously published genome. If the latter, please provide the relevant citation for the used published reference. Including this information in the Materials and Methods section would enhance the transparency of the study's methodology.
5- In the Materials and Methods section, it is essential to include comprehensive details regarding the utilized information, including default parameters of employed software. This should encompass specifics such as Gene IDs of the sequences, especially for reference genes in expression analysis, gap penalty, gap extension values, and the exact version of software. If online tools were used, please provide the access date. Providing this detailed information will enhance the reproducibility and transparency of the study's methodology
6- The arrowheads in Figure 7 appear to be improperly included, leading to potential confusion. I recommend a careful revision of the figure to ensure accurate representation.
7- The Neighbor-Joining phylogenetic tree would benefit from the inclusion of HSP genes from other orchid species. Integrating a broader range of orchid HSP sequences could provide a more comprehensive perspective on the evolutionary relationships within this gene family. Considering the relevance of comparative analysis, including additional orchid species in the NJ phylogenetic tree would strengthen the study's insights into the evolutionary dynamics of HSP genes in orchids.
9- The manuscript needs moderate language polishing which could be made as a final check.
Despite the aforementioned comments, I rate the manuscript as strong and strongly recommend its publication after minor revisions to address the raised questions. The proposed revisions would further enhance the clarity and completeness of the study, contributing to its overall strength and scientific meriف
Author Response
Comments and Suggestions for Authors
The manuscript entitled “General analysis of CeHSF genes in the Cymbidium ensifolium genome provided insights into their evolution and specific roles in tissue development and response to temperature” identify CeHSF genes from previously published C. ensifolium genome and analyze their phylogeny, gene structure, motif composition, expression patterns, and cis-elements. In total the manuscript needs minor revision to be ready for further consideration. With comments:
1- The abstract effectively outlines the identification and roles of CeHSF genes in Cymbidium ensifolium. However, the absence of quantitative data limits its clarity and immediate impact. Integrating key quantitative findings into the abstract would enhance its overall effectiveness.
R: Thank you for your review and valuable suggestions. Your suggestions made a lot of sense, so we revised the abstract. We highlight our key data and findings in the abstract. We hope that this revision concurs your expectations.
2- While the manuscript offers valuable insights into Cymbidium ensifolium's heat shock factors (HSFs), there is a notable absence of background information on the species' biology and specific challenges, especially in hot regions and the problem of heat stress . Incorporating this context would enhance the manuscript's significance and make it more accessible to a broader readership.
R: Thank you for your review and valuable suggestions. Your suggestions made a lot of sense, so we revised the Introduction of manuscript. To enhance the reader's understanding of the manuscript's content and significance, we have supplemented information on the background and challenges related to Cymbidium ensifolium in hot regions and heat stress issues.
Line 43. we revised it as “In recent years, abiotic stresses such as drought, high temperatures, salinity, and rising carbon dioxide have affected plant growth and agricultural development[5-7]. Among these, high temperature is one of the biggest challenges faced worldwide[8]. In the past few decades, there has been an increase in regions experiencing extremely high temperatures in China. This trend is primarily concentrated in the Yangtze River Delta region, eastern Sichuan, the Beijing-Tianjin-Hebei region, and the southeastern coastal areas[9]. Fujian Province, as the origin of C. ensifolium, has experienced dramatic temperature changes and frequent heat events in recent decades[10]. Due to various abiotic stresses, the habitats of C. ensifolium are declining, leading to the shrinking wild populations[11, 12]. C. ensifolium currently domesticated as vital ornamental orchids, have formed a global industry chain, requiring large quantities to meet market demands. Gaining a clear understanding of the heat response mechanisms of C. ensifolium is one of the key methods in the modern orchid breeding. However, there has been a scarcity of relevant researches.”
3- It is crucial to acknowledge and cite several closely related works that appear relevant to the current study but are currently absent from the references. Incorporating these works would strengthen the manuscript by placing the findings within a broader context and acknowledging existing contributions to the field:
Li, X., Liu, L., Sun, S., Li, Y., Jia, L., Ye, S., ... & Luan, Y. (2023). Transcriptome analysis reveals the key pathways and candidate genes involved in salt stress responses in Cymbidium ensifolium leaves. BMC Plant Biology, 23(1), 64.
Wang, M. J., Ou, Y., Li, Z., Zheng, Q. D., Ke, Y. J., Lai, H. P., ... & Ai, Y. (2023). Genome-Wide Identification and Analysis of bHLH Transcription Factors Related to Anthocyanin Biosynthesis in Cymbidium ensifolium. International Journal of Molecular Sciences, 24(4), 3825.
R: Thank you for your review and valuable suggestions. We noticed that the references you provided were very valuable and suitable for inclusion in this manuscript. Therefore, we added these references to the manuscript.
4- The origin of the analyzed sequences is not explicitly stated in the manuscript. It would be beneficial to clarify whether the authors obtained these sequences through their own sequencing efforts or utilized a previously published genome. If the latter, please provide the relevant citation for the used published reference. Including this information in the Materials and Methods section would enhance the transparency of the study's methodology.
R: Thank you for your review and valuable suggestions. We have added the source of the original sequence to the manuscript.
Line 350. We added “The genomic sequence and annotation data for C. ensifolium[46] were downloaded from the National Genome Data Center (NGDC) (https://ngdc.cncb.ac.cn/, accessed on 22 Oct 2023). ”
[46] Ai, Y.; Li, Z.; Sun, W.-H.; Chen, J.; Zhang, D.; Ma, L.; Zhang, Q.-H.; Chen, M.-K.; Zheng, Q.-D.; Liu, J.-F.; Jiang, Y.-T.; Li, B.-J.; Liu, X.; Xu, X.-Y.; Yu, X.; Zheng, Y.; Liao, X.-Y.; Zhou, Z.; Wang, J.-Y.; Wang, Z.-W.; Xie, T.-X.; Ma, S.-H.; Zhou, J.; Ke, Y.-J.; Zhou, Y.-Z.; Lu, H.-C.; Liu, K.-W.; Yang, F.-X.; Zhu, G.-F.; Huang, L.; Peng, D.-H.; Chen, S.-P.; Lan, S.; Van de Peer, Y.; Liu, Z.-J., The Cymbidium genome reveals the evolution of unique morphological traits. Hortic Res 2021, 8, (1), 255.
5- In the Materials and Methods section, it is essential to include comprehensive details regarding the utilized information, including default parameters of employed software. This should encompass specifics such as Gene IDs of the sequences, especially for reference genes in expression analysis, gap penalty, gap extension values, and the exact version of software. If online tools were used, please provide the access date. Providing this detailed information will enhance the reproducibility and transparency of the study's methodology
R: Thank you for your review and valuable suggestions. Your suggestions made a lot of sense, so we revised the Materials and Methods section. We added the parameters of the software used and the access date of the online tools.
About the analysis of gene expression patterns section, reads were counted on transcript mapping analysis, this is a genome-free method with rapid operation and accurate results. The kallisto software was the version 0.48.0, with parameters default. References were as follows: [1, 2]
[1] Bray, N. L.; Pimentel, H.; Melsted, P.; Pachter, L., Near-optimal probabilistic RNA-seq quantification. Nat Biotechnol 2016, 34, (5), 525-527.
[2] Coughlin, D. J.; Nicastro, L. K.; Brookes, P. J.; Bradley, M. A.; Shuman, J. L.; Steirer, E. R.; Mistry, H. L., Thermal acclimation and gene expression in rainbow smelt: Changes in the myotomal transcriptome in the cold. Comp Biochem Physiol Part D Genomics Proteomics 2019, 31, 100610.
6- The arrowheads in Figure 7 appear to be improperly included, leading to potential confusion. I recommend a careful revision of the figure to ensure accurate representation.
R: Thank you for your review and valuable suggestions. Your suggestions made a lot of sense, so we revised the Figure 7. We hope that this revision concurs your expectations.
7- The Neighbor-Joining phylogenetic tree would benefit from the inclusion of HSP genes from other orchid species. Integrating a broader range of orchid HSP sequences could provide a more comprehensive perspective on the evolutionary relationships within this gene family. Considering the relevance of comparative analysis, including additional orchid species in the NJ phylogenetic tree would strengthen the study's insights into the evolutionary dynamics of HSP genes in orchids.
R: Thank you for your review and valuable suggestions. Our research focused on the Cymbidium ensifolium genome, specifically investigating the evolution and function of Heat Shock Factor (HSF) genes within Cymbidium ensifolium. Consequently, we exclusively considered the HSF genes in Cymbidium ensifolium. However, your suggestion was highly meaningful and provided significant inspiration. In our next article, we plan to systematically analyze the evolutionary relationships of HSF genes in the Orchidaceae plant.
9- The manuscript needs moderate language polishing which could be made as a final check.
R: Thank you for your review and valuable suggestions. Your suggestions made a lot of sense, so we revised the entire manuscript. We corrected some mistakes in words and grammar. We hope that this revision concurs your expectations.
Despite the aforementioned comments, I rate the manuscript as strong and strongly recommend its publication after minor revisions to address the raised questions. The proposed revisions would further enhance the clarity and completeness of the study, contributing to its overall strength and scientific meriف
R: Thank you for your review and valuable suggestions. Please do not hesitate to contact me if you have any other questions.
Reviewer 3 Report
Comments and Suggestions for Authors
The reviewed work concerned the analysis of CeHSF genes in the genome of Cymbidium ensifolium.
The topic that the authors dealt with is very interesting. In the paper, the authors emphasized the novelty of the study and justified its purpose well.
The results are described in great detail and are presented in tables and figures. The authors have also included additional tables and figures.
The research methods are described correctly.
Discussion - in my opinion is too short. The authors should discuss in detail the problem addressed in the paper and present the results of other researchers.
The summary is understandable and sufficient.
The authors cited 52 items of literature. Among them, 31 were works from the last 5 years.
Comment:
1. Please improve the discussion.
Author Response
Comments and Suggestions for Authors
The reviewed work concerned the analysis of CeHSF genes in the genome of Cymbidium ensifolium.
The topic that the authors dealt with is very interesting. In the paper, the authors emphasized the novelty of the study and justified its purpose well.
The results are described in great detail and are presented in tables and figures. The authors have also included additional tables and figures.
The research methods are described correctly.
Discussion - in my opinion is too short. The authors should discuss in detail the problem addressed in the paper and present the results of other researchers.
The summary is understandable and sufficient.
The authors cited 52 items of literature. Among them, 31 were works from the last 5 years.
Comment:
- Please improve the discussion.
R: Thank you for your review and valuable suggestions. Your suggestions made a lot of sense, so we revised the discussion of manuscript. We supplemented the discussion with other results, such as miRNA analysis and Protein-protein interaction network analysis. This enriched the content of the discussion section in our manuscript. We hope that this revision concurs your expectations.
Round 2
Reviewer 1 Report
Comments and Suggestions for Authors
No suggestion!